# Towards stronger antenatal care: Understanding predictors of late presentation to antenatal services and implications for obstetric risk management in Rwanda

Christina N. Schmidt[1]*, Elizabeth Butrick[2], Sabine Musange[3], Nathalie Mulindahabi[3], Dilys Walker[2,4]

1 School of Medicine, University of California San Francisco, San Francisco, California, United States of America, 2 Institute for Global Health Sciences, University of California San Francisco, San Francisco, California, United States of America, 3 School of Public Health, National University of Rwanda, Kigali, Rwanda, 4 Department of Obstetrics, Gynecology & Reproductive Sciences, University of California San Francisco, San Francisco, California, United States of America

* Christina.Schmidt@ucsf.edu

**Data Availability Statement:** Data files are available at the following DOI: 10.17605/OSF.IO/NKABY.

## Abstract

### Background

Early antenatal care (ANC) reduces maternal and neonatal morbidity and mortality through identification of pregnancy-related complications, yet 44% of Rwandan women present to ANC after 16 weeks gestational age (GA). The objective of this study was to identify factors associated with delayed initiation of ANC and describe differences in the obstetric risks identified at the first ANC visit (ANC-1) between women presenting early and late to care.

### Methods

This secondary data analysis included 10,231 women presenting for ANC-1 across 18 health centers in Rwanda (May 2017-December 2018). Multivariable logistic regression models were constructed using backwards elimination to identify predictors of presentation to ANC at $\geq$16 and $\geq$24 weeks GA. Logistic regression was used to examine differences in obstetric risk factors identified at ANC-1 between women presenting before and after 16- and 24-weeks GA.

### Results

Sixty-one percent of women presented to ANC at $\geq$16 weeks and 24.7% at $\geq$24 weeks GA, with a mean (SD) GA at presentation of 18.9 (6.9) weeks. Younger age (16 weeks: OR = 1.36, 95% CI: 1.06, 1.75; 24 weeks: OR = 1.33, 95% CI: 0.95, 1.85), higher parity (16 weeks: 1–4 births, OR = 1.55, 95% CI: 1.39, 1.72; five or more births, OR = 2.57, 95% CI: 2.17, 3.04; 24 weeks: 1–4 births, OR = 1.93, 95% CI: 1.78, 2.09; five or more births, OR = 3.20, 95% CI: 2.66, 3.85), lower educational attainment (16 weeks: primary, OR = 0.75, 95% CI: 0.65, 0.86; secondary, OR = 0.60, 95% CI: 0.47,0.76; university, OR = 0.48, 95% CI: 0.33, 0.70; 24 weeks: primary, OR = 0.64, 95% CI: 0.53, 0.77; secondary, OR = 0.43,

**Funding:** This work is part of the East Africa Preterm Birth Initiative which is funded by the Bill & Melinda Gates Foundation. The funders had no role in study design, data collection and analysis, decision to publish, or preparation of the manuscript.

**Competing interests:** The authors have declared that no competing interests exist.

95% CI: 0.29, 0.63; university, OR = 0.12, 95% CI: 0.04, 0.32) and contributing to household income (16 weeks: OR = 1.78, 95% CI: 1.40, 2.25; 24 weeks: OR = 1.91, 95% CI: 1.42, 2.55) were associated with delayed ANC-1 ($\geq$16 and $\geq$24 weeks GA). History of a spontaneous abortion (16 weeks: OR = 0.74, 95% CI: 0.66, 0.84; 24 weeks: OR = 0.70, 95% CI: 0.58, 0.84), pregnancy testing (16 weeks: OR = 0.48, 95% CI: 0.33, 0.71; 24 weeks: OR = 0.41, 95% CI: 0.27, 0.61; 24 weeks) and residing in the same district (16 weeks: OR = 1.55, 95% CI: 1.08, 2.22; 24 weeks: OR = 1.73, 95% CI: 1.04, 2.87) or catchment area (16 weeks: OR = 1.53, 95% CI: 1.05, 2.23; 24 weeks: OR = 1.84, 95% CI: 1.28, 2.66; 24 weeks) as the health facility were protective against delayed ANC-1. Women with a prior preterm (OR, 0.71, 95% CI, 0.53, 0.95) or low birthweight delivery (OR, 0.72, 95% CI, 0.55, 0.95) were less likely to initiate ANC after 16 weeks. Women with no obstetric history were more likely to present after 16 weeks GA (OR, 1.18, 95% CI, 1.06, 1.32).

## Conclusion

This study identified multiple predictors of delayed ANC-1. Focusing existing Community Health Worker outreach efforts on the populations at greatest risk of delaying care and expanding access to home pregnancy testing may improve early care attendance. While women presenting late to care were less likely to present without an identified obstetric risk factor, lower than expected rates were identified in the study population overall. Health centers may benefit from provider training and standardized screening protocols to improve identification of obstetric risk factors at ANC-1.

## Introduction

Attendance at antenatal care (ANC) is important for the health outcomes of expectant mothers and future neonates. Through early identification, management and referral of pregnancy-related complications, ANC can contribute to reductions in maternal and neonatal morbidity and mortality [1–5]. While global maternal and neonatal mortality rates fell by 44% [6] and 47% [7] between 1990 and 2015, inequities persist across geographic regions. Sub-Saharan Africa maintains the highest regional rate of maternal mortality, accounting for more than half of maternal deaths globally [6]. The region also has the highest neonatal mortality rate, comprising 41% of neonatal deaths worldwide [8].

Many obstetric complications leading to adverse outcomes for mothers and neonates can be effectively managed if identified at an ANC visit early in pregnancy. Leading causes of maternal mortality in sub-Saharan Africa, including hypertensive disorders and pregnancy-related infections, can be mitigated in the antenatal period resulting in lower mortality rates [9, 10]. Decreased rates of preterm birth, a common cause of neonatal morbidity and mortality in low-resources settings, have been linked to increased attendance at ANC [11]. Surveillance throughout the antenatal period for conditions such as diabetes, anemia and HIV can further reduce obstetric complications through early monitoring and management [12, 13]. Attendance at ANC may also increase the usage of emergency obstetric care [14], and motivate women to deliver at a health facility [15]. Given the known benefits to mothers and neonates, Rwanda has made increasing attendance at ANC a national priority [16].

Despite recognition of the importance of early ANC attendance, many pregnant women in resource-poor settings continue to present late to care. In Rwanda, women are encouraged to

present to ANC before 16 weeks GA and attend four prenatal visits, yet compliance with ANC schedules is low. The 2015 Rwandan Demographic and Health Survey found that only 56% of women present to ANC before 16 weeks GA [16]. This is consistent with literature from the sub-Saharan Africa region, which has frequently reported high rates of late presentation to ANC [16–18]. While 99% of pregnant women in Rwanda attend at least one ANC visit prior to delivery, only 52% attend 2 or 3 visits and only 44% attend the Rwandan Ministry of Health's recommended four ANC visits [16].

Various predictors of delayed initiation of ANC have been identified in low-resource countries. These include lower educational attainment, decreased household income, higher cost of care and unemployment [19]. In sub-Saharan Africa, unplanned pregnancy, increased distance from a health facility and food insecurity have also been associated with late presentation to ANC [4, 16, 19–23]. The financial burden of seeking care, completing demands on family resources, and long travel times may make accessing care a challenge for families. In Rwanda, cultural norms, such as the practice of disclosing pregnancy after the second trimester, when the pregnancy is externally visible, have been suggested as a reason for delayed ANC initiation [24]. Negative experiences with ANC providers, including receiving criticism for registering for care either too early or too late, or arriving without accompaniment by a male partner, have also been reported as barriers to attending ANC in Rwanda [24].

Some evidence also suggests that women present late to ANC because they believe that their pregnancies are low-risk [24–26]. This may be especially true among multiparous women with a history of uncomplicated births [21, 26, 27]. Various studies have shown that increased parity and a history of uncomplicated deliveries are associated with delayed ANC [19, 20], which may be due to reassurance from previous uncomplicated pregnancies. Women with uncomplicated prior pregnancies may actually be at lower risk of developing certain obstetric complications compared to their nulliparous counterparts. The strongest predictor of complications such as preterm birth and pre-eclampsia, for example, is a history of these conditions in a prior pregnancy [27–29].

Improving ANC attendance has become increasingly relevant in the wake of the World Health Organization's 2016 release of clinical guidelines recommending that women have at least eight contacts with the health system throughout their pregnancies [12]. Previously, the World Health Organization's focused antenatal care (FANC) strategy encouraged four ANC visits during the prenatal period [30, 31]. While the World Health Organization asserts that increasing the number of ANC contacts leads to improved health outcomes, some data suggest that the quality and content of ANC care, rather than the number of contacts, has a greater impact on maternal and neonatal outcomes [31–33]. Results from one systematic review suggest that women with low-risk pregnancies can safely have fewer ANC visits [34]. As international recommendations shift towards increasing the number of ANC contacts, understanding predictors of late presentation to care, as well as the impact of late presentation on clinically significant outcomes is increasingly relevant.

This analysis identified predictors of late (≥16 weeks GA) and very late (≥24 weeks GA) presentation to ANC among a cohort of 10,231 women enrolled in a randomized controlled trial on prenatal care and birth outcomes in Rwanda (The Preterm Birth Initiative–Rwanda) [35]. Through understanding predictors of delayed presentation to care, and identifying which factors are most amenable to change, policymakers can more effectively develop strategies to increase earlier attendance at ANC visits. This analysis also assessed differences in the pregnancy-related risk factors identified at a woman's first ANC visit, based on both parity and GA at presentation to care. Understanding the differences in risk profiles between pregnant women who present early and those who present late to ANC will contribute to how we

understand the risks of late presentation and inform messages tailored to multiparous and nulliparous women.

## Methods

### Study design and population

This secondary data analysis used data obtained in a cluster randomized controlled trial on prenatal care and birth outcomes, conducted by The Preterm Birth Initiative–Rwanda (NCT03154177) [35]. The present analysis was restricted to health centers randomized to the control group, which included 18 facilities in 5 districts (Bugesera, Rubavu, Nyamasheke, Nyarugenge, and Burera). These health centers were selected for inclusion in the trial based on their location in one of the five districts, their monthly ANC volume, and the presence of at least two ANC providers at the facility. Women ≥15 years of age who presented to one of the participating health centers between May 2017 and December 2018 for ANC services were invited to enroll in the study. Only participants with completed ANC records were included in the final analysis. There were no significant differences in the characteristics of those with incomplete ANC records, as determined by linear models of key sociodemographic variables.

### Enrollment surveys

Trained data collectors employed by the research team were embedded at each of the 18 health centers. After introducing the study to each woman presenting for ANC, data collectors obtained consent for participants to be included in the study. Upon enrollment, data collectors administered an initial survey to all participants (S1 File). Data collected included age, educational attainment, occupation, contribution to household income, level of partner communication, proximity of the health center to their home and tobacco and alcohol use. Food security over the past month was assessed with a two-question series recommended by the American Association of Pediatrics [36]. Women were also asked whether they had received a pregnancy test and/or whether a community health worker (CHW) had recommended that they visit a health center to confirm their pregnancy.

Multiparous participants were asked to report any previous preterm births, low birth weight infants, fresh stillbirths, neonatal deaths (first 28 days of life) and repeated miscarriages. These participants were also asked to report the number of ANC appointments that they attended during their most recent past pregnancy.

### Antenatal visit data

Data from participants' first ANC visit (ANC-1) were abstracted from existing national collection tools, including health center registers and patient files. All health centers participated in data strengthening training prior to the start of the study to improve accuracy and completeness of these existing data collection tools. Information abstracted from participants' antenatal registers included gravidity, parity, and GA at ANC-1. Obstetric risk factors included the presence of anemia, proteinuria, hypertension (≥140/90), multiple births, middle upper arm circumference (MUAC) <21cm, and HIV positive status (either positive test or known positive status documented in the chart). Syphilis or malaria identified at ANC-1 were also recorded. If no obstetric risk factors were identified in the chart, data collectors recorded "none." Additional history collected from participants' ANC-1 files included a documented history of diabetes and/or chronic hypertension. In Rwanda, anemia, proteinuria, hypertension, multiple births, MUAC <21cm, diabetes, and syphilis are universally screened for at ANC-1 and noted as either present or absent in the maternity register using tick boxes.

## Analysis

The primary outcomes in this analysis were late and very late presentation to ANC. Late presentation was defined as $\geq$16 weeks GA in accordance with the Rwandan Ministry of Health's prenatal care performance indicators, and very late was defined as $\geq$24 weeks GA [37]. Univariate logistic regression models were constructed to identify variables significantly associated with late presentation to ANC at both GA cut-offs for all 10,231 participants, both nulliparous and multiparous. Variables assessed included age, educational attainment, employment, contribution to household income, food insecurity, health center proximity, prior spontaneous abortion, gravidity/parity, pregnancy testing, CHW recommended health center visit, partner communication and the health center proximity.

Variables that were significantly associated with delayed presentation to ANC at the $\alpha_{crit}$ = 0.20 level in univariate analyses were retained for multivariable model building [38, 39]. For covariates that were identified to be collinear (variance inflation factor >2.5) the variable more strongly associated with delayed ANC was retained. Final multivariable logistic regression models were constructed using manual backwards elimination. A full model including all candidate predictors was constructed, and the predictor with the highest p-value greater than $\alpha_{crit}$ = 0.20 was removed. The model was refit and this process was repeated until all variables maintained in the model had a p-value less than $\alpha_{crit}$, with the exception of age which was considered by the investigators to be a potential confounder. Cluster-robust standard errors were used to account for the clustering effects of health centers. Odds ratios and 95% confidence intervals are reported.

For the 7,380 multiparous participants, obstetrics history predictors of late ($\geq$16 weeks GA) and very late ($\geq$24 weeks GA) presentation to ANC were assessed using logistic regression models. Obstetric history variables included a history of a preterm delivery, low birthweight infant, previous fresh stillbirth, 28-day mortality of a neonate, and repeated miscarriages. Self-reported ANC attendance in the most recent prior pregnancy was also assessed.

The secondary outcomes of interest were the obstetric risk factors identified at ANC-1. Logistic regression models were used to identify associations between late and very late presentation to ANC and the types of obstetric risk factors identified at a woman's first ANC visit. Risk factors assessed included anemia, proteinuria, hypertension, multiple births, smoking, alcohol use, HIV positive status, and MUAC <21. Diabetes, syphilis and malaria were not reliably recorded in the ANC-1 records, and thus were excluded from the final analysis.

Additional logistic regression models were also used to assess for associations between parity and the identification of pregnancy-related risk factors at ANC-1. To assess whether parity was a moderator of the relationship between GA at ANC-1 and each of the obstetric risk factors identified at ANC-1, logistic regression models with interaction terms were used. All analyses were conducted in R (version 3.6.1).

## Ethical considerations

This study was approved by the Rwanda National Ethics Committee (No.0034/RNEC/2017), and the University of California, San Francisco Institutional Review Board (16–21177). Written consent was obtained from all participants prior to administering the enrollment survey and reviewing patient health records. All consent forms were translated into Kinyarwanda. Participants provided consent by reading and signing the consent form. For participants who were illiterate, a member of the study team verbally read the consent form in the presence of a witness and both the consented participant and witness signed the consent. The Rwandan National Ethics Committee and the University of California, San Francisco Institutional Review Board waived parental consent requirements for pregnant minors.

## Results

### Predictors of delayed ANC

Data were analyzed for 10,231 eligible women presenting to ANC-1 at participating health centers. The majority of women (72.3%) were between the ages of 20–35 (mean 28.9 years) and most (53.9%) had not completed primary school. Sixty one percent of women presented to ANC at ≥16 weeks GA, and 25% presented to ANC at ≥24 weeks GA, with an mean (SD) GA at presentation of 18.9 (6.9) weeks. Seventy two percent of the women in the cohort were multiparous, and of these multiparous women 37.5% had attended ≥4 ANC visits during their most recent prior pregnancy.

Several factors were significantly associated with late and very late presentation to ANC in univariate models (Table 1). Age (p<0.001), lower educational attainment (p<0.001), contribution to household income (p<0.001), gravity (p<0.001), parity (p<0.001), and recommendation from a CHW to visit a health center to confirm pregnancy (p<0.001) were significantly associated with presentation to ANC at ≥16 weeks and ≥24 weeks GA. History of a spontaneous abortion (16 weeks, p<0.003; 24 weeks, p = 0.005), and pregnancy testing (p<0.001) were protective against late presentation to ANC. Food insecurity was associated with presentation to ANC at ≥24 weeks GA (p<0.001), and decreased communication between partners was associated with presentation at ANC ≥16 weeks GA (p = 0.001). There were significant differences in the rate of late presentation (≥16 and ≥ 24 weeks GA) between districts (p<0.001).

The association between having a history of an obstetric complication and GA at ANC-1 among multiparous women is reported in Table 2. Women with no obstetric history were more likely to present at ≥16 weeks GA (OR, 1.18, 95% CI, 1.06, 1.32), while women with a history of a preterm delivery (OR, 0.71, 95% CI, 0.53, 0.95) and low birthweight baby (OR, 0.72, 95% CI, 0.55, 0.95) were less likely to delay care beyond 16 weeks GA. Other obstetric risk factors (fresh stillbirth, 28-day neonatal mortality and repeated miscarriage) were not associated with timing of ANC-1. Attending ≥4 ANC visit in the most recent prior pregnancy was associated with a decreased risk of delayed ANC-1 (16 weeks: OR, 0.66, 95% CI, 0.62, 0.70; 24 weeks: OR, 0.61, 95% CI, 0.57, 0.65).

Both full and reduced multivariable models for late (≥16 weeks GA) and very late (≥24 weeks GA) presentation to ANC are shown in Table 3. Factors significantly associated with late presentation to ANC included increased parity (1–4 births, OR = 1.55, 95% CI: 1.39, 1.72; five or more births, OR = 2.57, 95% CI: 2.17, 3.04), lower educational attainment (primary, OR = 0.75, 95% CI: 0.65, 0.86; secondary, OR = 0.60, 95% CI: 0.47,0.76; university, OR = 0.48, 95% CI: 0.33, 0.70), contributing to household income (OR = 1.78, 95% CI: 1.40, 2.25) and residing in the same district (OR = 1.55, 95% CI: 1.08, 2.22) or catchment area (OR = 1.53, 95% CI: 1.05, 2.23) as the health facility. These same factors–increased parity (1–4 births, OR = 1.93, 95% CI: 1.78, 2.09; five or more births, OR = 3.20, 95% CI: 2.66, 3.85), lower educational attainment (primary, OR = 0.64, 95% CI: 0.53, 0.77; secondary, OR = 0.43, 95% CI: 0.29, 0.63; university, OR = 0.12, 95% CI: 0.04, 0.32), contributing to household income (OR = 1.91, 95% CI: 1.42, 2.55) and residing in the same district (OR = 1.73, 95% CI: 1.04, 2.87) or catchment area (OR = 1.84, 95% CI: 1.28, 2.26) as the health facility–were also associated with very late presentation to care. History of a spontaneous abortion (16 weeks: OR = 0.74, 95% CI: 0.66, 0.84; 24 weeks: OR = 0.70, 95% CI: 0.58, 0.84) and receiving a pregnancy test (16 weeks: OR = 0.48, 95% CI: 0.33, 0.71; 24 weeks: OR = 0.41, 95% CI: 0.27, 0.61) were protective against late and very late presentation to ANC. Younger age (15–19 years) predicted presentation to ANC at ≥16 weeks GA (OR = 1.36, 95% CI: 1.06, 1.75) and at ≥24 weeks GA (OR = 1.33, 95% CI: 0.95, 1.85), although the latter was not statistically significant. Employment predicted

**Table 1. Univariate relationships between sociodemographic factors and delayed antenatal care.**

| Characteristic | ≥ 16 weeks gestational age | | | | | ≥ 24 weeks gestational age | | | | |
|---|---|---|---|---|---|---|---|---|---|---|
| | N | % | OR | 95% CI | P-value | N | % | OR | 95% CI | P-value |
| **Age** (ref = 25–29 years) | | | | | <0.001 | | | | | <0.001 |
| 15–19 | 477 | 6.2 | 1.04 | 0.87, 1.25 | | 119 | 4.7 | 0.83 | 0.67, 1.03 | |
| 20–24 | 1862 | 24.2 | 0.95 | 0.85, 1.06 | | 514 | 20.3 | 0.92 | 0.81, 1.05 | |
| 25–29 | 2091 | 27.1 | 1.00 | (ref) | | 626 | 24.8 | 1.00 | (ref) | |
| 30–34 | 1698 | 22.0 | 1.19 | 1.06, 1.33 | | 609 | 24.1 | 1.20 | 1.05, 1.36 | |
| 35+ | 1577 | 20.5 | 1.29 | 1.15, 1.45 | | 658 | 26.0 | 1.39 | 1.23, 1.58 | |
| **Gravidity** (ref = 1) | | | | | <0.001 | | | | | <0.001 |
| 1 | 1430 | 22.9 | 1.00 | (ref) | | 441 | 17.5 | 1.00 | (ref) | |
| 2–4 | 3130 | 50.1 | 1.38 | 1.25, 1.51 | | 1275 | 50.5 | 1.68 | 1.49, 1.90 | |
| 5+ | 1686 | 27.0 | 1.88 | 1.67, 2.10 | | 810 | 32.1 | 2.47 | 2.16, 2.82 | |
| **Parity** (ref = 0) | | | | | <0.001 | | | | | <0.001 |
| 0 | 1522 | 24.4 | 1.00 | (ref) | | 459 | 18.2 | 1.00 | (ref) | |
| 1–4 | 3809 | 61.0 | 1.45 | 1.32, 1.59 | | 1609 | 63.7 | 1.87 | 1.67, 2.01 | |
| 5+ | 915 | 14.6 | 2.18 | 1.89, 2.52 | | 458 | 18.1 | 2.90 | 2.49, 3.38 | |
| **Educational attainment** (ref = none) | | | | | <0.001 | | | | | <0.001 |
| None | 3578 | 57.3 | 1.00 | (ref) | | 1621 | 64.2 | 1.00 | (ref) | |
| Primary | 2168 | 34.7 | 0.74 | 0.68, 0.81 | | 773 | 30.6 | 0.62 | 0.56, 0,69 | |
| Secondary | 450 | 7.2 | 0.62 | 0.53, 0.72 | | 127 | 5.0 | 0.43 | 0.35, 0.52 | |
| University | 50 | 0.8 | 0.52 | 0.35, 0.77 | | 5 | 0.2 | 0.12 | 0.04, 0.28 | |
| **Employed** (ref = no) | 5678 | 90.9 | 0.93 | 0.81, 1.07 | 0.309 | 2321 | 91.9 | 1.13 | 0.97, 1.34 | 0.127 |
| **Contributes to household income** (ref = no) | 1499 | 24.0 | 1.53 | 1.38, 1.69 | <0.001 | 700 | 27.7 | 1.61 | 1.45, 1.79 | <0.001 |
| **Food insecurity** (ref = no) | 3942 | 63.1 | 0.93 | 0.86, 1.02 | 0.115 | 1687 | 66.8 | 1.20 | 1.09, 1.31 | <0.001 |
| **Prior spontaneous abortion** (ref = no) | 651 | 10.4 | 0.83 | 0.73, 0.94 | 0.003 | 243 | 9.6 | 0.81 | 0.69, 0.93 | 0.005 |
| **Pregnancy test** (ref = no) | 357 | 5.7 | 0.49 | 0.42, 0.56 | <0.001 | 102 | 4.0 | 0.42 | 0.34, 0.52 | <0.001 |
| **CHW recommended health center visit** (ref = no) | 1830 | 29.3 | 0.86 | 0.79, 0.94 | <0.001 | 690 | 27.3 | 0.81 | 0.74, 0.90 | <0.001 |
| **Open communication with partner** (ref = no) | 5643 | 90.3 | 0.79 | 0.69, 0.91 | 0.001 | 2299 | 91.0 | 0.99 | 0.85, 1.16 | 0.914 |
| **Health center proximity** (ref = resides outside of district) | | | | | 0.486 | | | | | <0.001 |
| Resides outside of district | 54 | 0.9 | 1.00 | (ref) | | 16 | 0.6 | 1.00 | (ref) | |
| Resides within district | 848 | 13.6 | 1.37 | 0.92, 2.02 | | 330 | 13.1 | 1.76 | 1.06, 3.13 | |
| Resides within catchment area | 5344 | 85.6 | 1.38 | 0.90, 2.07 | | 2180 | 86.3 | 1.67 | 0.99, 2.98 | |
| **District** (ref = Rubavu) | | | | | <0.001 | | | | | 0.035 |
| Bugesera | 1645 | 26.3 | 0.30 | 0.27, 0.34 | | 481 | 19.0 | 0.19 | 0.17, 0.21 | |
| Burera | 1110 | 17.8 | 0.25 | 0.21, 0.28 | | 347 | 13.7 | 0.18 | 0.16, 0.21 | |
| Nyarugenge | 769 | 12.3 | 0.18 | 0.15, 0.20 | | 234 | 9.3 | 0.15 | 0.12, 0.17 | |
| Nyamasheke | 793 | 12.7 | 0.35 | 0.30, 0.41 | | 241 | 9.5 | 0.21 | 0.18, 0.25 | |
| Rubavu | 2698 | 43.2 | 1.00 | (ref) | | 1457 | 57.7 | 1.00 | (ref) | |

presentation to care at ≥16 weeks GA (OR = 0.64, 95% CI: 0.51, 0.82) and at ≥24 weeks GA (OR = 0.69, 95% CI: 0.46, 1.04), although again the latter was not statistically significant.

## Obstetric risk factors identified at ANC-1

Analyses of obstetric risk factors identified at ANC-1 revealed differences in documented risk factors based on both GA at presentation to care and parity (Table 4). Women presenting to care at ≥16 and ≥24 weeks GA were more likely to have multiple births (16 weeks: OR = 1.90, 95% CI: 1.47, 2.50; 24 weeks: OR = 1.74, 95% CI: 1.36, 2.21) and a history of alcohol use (16

**Table 2. Univariate relationships between obstetric history and delayed antenatal care.**

| Obstetric history | ≥ 16 weeks gestational age | | | | | ≥ 24 weeks gestational age | | | | |
|---|---|---|---|---|---|---|---|---|---|---|
| | N | % | OR | 95% CI | P-value | N | % | OR | 95% CI | P-value |
| None | 3688 | 78.1 | 1.18 | 1.06, 1.32 | 0.003 | 1614 | 78.1 | 1.09 | 0.97, 1.23 | 0.164 |
| Preterm delivery | 107 | 2.3 | 0.71 | 0.53, 0.95 | 0.020 | 42 | 2.0 | 0.72 | 0.50, 1.01 | 0.062 |
| Low birthweight infant | 125 | 2.6 | 0.72 | 0.55, 0.95 | 0.020 | 59 | 2.9 | 0.93 | 0.69, 1.26 | 0.659 |
| Previous fresh stillbirth | 220 | 4.7 | 0.96 | 0.77, 1.20 | 0.698 | 95 | 4.6 | 0.96 | 0.75, 1.22 | 0.737 |
| 28-day mortality of a neonate | 118 | 2.5 | 1.04 | 0.77, 1.42 | 0.815 | 54 | 2.6 | 1.09 | 0.78, 1.49 | 0.613 |
| Repeated miscarriage | 118 | 2.5 | 0.86 | 0.64, 1.15 | 0.303 | 48 | 2.3 | 0.84 | 0.60, 1.15 | 0.286 |
| ≥4 ANC visits in last pregnancy | 1511 | 32.0 | 0.66 | 0.62, 0.70 | <0.001 | 511 | 24 | 0.61 | 0.57, 0.65 | <0.001 |

weeks: OR = 1.16, 95% CI: 1.04, 1.29; 24 weeks: OR = 1.15, 95% CI: 1.01, 1.29). Patients who presented at <16 and <24 weeks GA were more likely to have proteinuria at ANC-1 (16 weeks: OR = 0.62, 95% CI: 0.56, 0.68; 24 weeks: OR = 0.62, 95% CI: 0.55, 0.72). Overall, women presenting to care at <16 weeks GA were more likely to have no obstetric risk factors identified and documented at ANC-1 (OR = 0.74, 95% CI: 0.63, 0.86) (Table 4).

Differences in obstetric risk factors identified at first ANC visit based on parity are reported in Table 5. Multiparous women (n = 7380) were more likely to have multiple births (OR = 58.9, 95% CI: 18.9, 335) and report alcohol use (OR = 1.68, 95% CI: 1.48, 1.93) at their first ANC visit. Women who were nulliparous were more likely to have no obstetric risk factors documented at ANC-1 (OR = 0.28, 95% CI: 0.22, 0.35). Parity was not a moderator of the relationship between GA at presentation to care and any of the obstetric risk factors identified at ANC-1 in our moderator analyses.

## Discussion

Among our cohort of 10,231 pregnant women, we found that three-fifths presented to ANC after 16 weeks GA and one-fourth presented after 24 weeks GA, with a mean GA at presentation of 19 weeks. These rates are similar to those reported by the Rwandan Ministry of Health [16], and are consistent with patterns of delayed initiation of ANC across East Africa [16–18]. While Rwanda has been successful in achieving near-universal attendance at ANC, our results demonstrate that the majority of women continue to present well beyond the country's goal of initiating care before 16 weeks GA. In order to realize the full benefits of ANC attendance–including early identification and management of pregnancy-related risk factors–increased emphasis should be placed on promoting early initiation of care.

Rwanda has made it a national priority to improve access to ANC. In 2007 Rwanda implemented a CHW outreach program as a primary strategy to promote ANC attendance [40]. CHWs are responsible for identifying pregnant women, providing prenatal health education, and encouraging attendance at ANC. Interventions that build on the existing outreach systems to increase early access to ANC will be most effective if they are tailored towards the populations at the greatest risk of delaying care. In this investigation we identified multiple predictors of late presentation to ANC-1, suggesting a few priority groups.

We found that lower educational attainment was associated with delayed ANC-1, which is consistent with findings from studies in the region [19, 22]. This might be directly due to increased knowledge about the importance of ANC or may be due to increased autonomy and financial independence associated with higher educational attainment. While both older and younger ages have been identified as predictors of late ANC initiation in other studies, we found that younger participants were more likely to delay ANC-1 in our cohort [17, 41, 42].

**Table 3. Multivariable predictors of late and very late presentation to antenatal care.**

| Characteristic | ≥ 16 weeks gestational age | | | | | | ≥ 24 weeks gestational age | | | | | |
|---|---|---|---|---|---|---|---|---|---|---|---|---|
| | Full | | | Reduced | | | Full | | | Reduced | | |
| | OR | 95% CI | P-value | OR | 95% CI | P-value | OR | 95% CI | P-value | OR | 95% CI | P-value |
| **Age** (ref = 25–29 years) | | | | | | | | | | | | |
| 15–19 | 1.36 | 1.06,1.73 | 0.014 | 1.36 | 1.06,1.75 | 0.016 | 1.33 | 0.97,1.83 | 0.080 | 1.33 | 0.95,1.85 | 0.094 |
| 20–24 | 1.13 | 0.95,1.35 | 0.178 | 1.13 | 0.95,1.35 | 0.160 | 1.21 | 1.03,1.42 | 0.020 | 1.20 | 1.03,1.41 | 0.020 |
| 25–29 | 1.00 | (ref) | | 1.00 | (ref) | | 1.00 | (ref) | | 1.00 | (ref) | |
| 30–34 | 1.04 | 0.92,1.19 | 0.505 | 1.04 | 0.92,1.19 | 0.506 | 0.99 | 0.85,1.17 | 0.950 | 0.99 | 0.85,1.17 | 0.943 |
| 35+ | 0.94 | 0.82,1.08 | 0.376 | 0.94 | 0.82,1.07 | 0.365 | 0.94 | 0.82,1.08 | 0.389 | 0.94 | 0.82,1.08 | 0.368 |
| **Parity** (ref = 0) | | | | | | | | | | | | |
| 0 | 1.00 | (ref) | | 1.00 | (ref) | | 1.00 | (ref) | | 1.00 | (ref) | |
| 1–4 | 1.56 | 1.39,1.74 | <0.001 | 1.55 | 1.39,1.72 | <0.001 | 1.91 | 1.74,2.10 | <0.001 | 1.93 | 1.78,2.09 | <0.001 |
| 5+ | 2.60 | 2.20,3.08 | <0.001 | 2.57 | 2.17,3.04 | <0.001 | 3.15 | 1.82,5.74 | <0.001 | 3.20 | 2.66,3.85 | <0.001 |
| **Educational attainment** (re = none) | | | | | | | | | | | | |
| None | 1.00 | (ref) | | 1.00 | (ref) | | 1.00 | (ref) | | 1.00 | (ref) | |
| Primary | 0.74 | 0.64,0.85 | <0.001 | 0.75 | 0.65,0.86 | <0.001 | 0.65 | 0.54,0.77 | <0.001 | 0.64 | 0.53,0.77 | <0.001 |
| Secondary | 0.58 | 0.46,0.75 | <0.001 | 0.60 | 0.47,0.76 | <0.001 | 0.44 | 0.30,0.63 | <0.001 | 0.43 | 0.29,0.63 | <0.001 |
| University | 0.46 | 0.31,0.69 | <0.001 | 0.48 | 0.33,0.70 | <0.001 | 0.12 | 0.04,0.35 | <0.001 | 0.12 | 0.04,0.32 | <0.001 |
| **Employed** (ref = no) | | | | | | | | | | | | |
| No | 1.00 | (ref) | | 1.00 | (ref) | | 1.00 | (ref) | | 1.00 | (ref) | |
| Yes | 0.65 | 0.51,0.84 | <0.001 | 0.64 | 0.51,0.82 | <0.001 | 0.68 | 0.47,0.99 | 0.046 | 0.69 | 0.46,1.04 | 0.077 |
| **Contributes to household income** (ref = no) | | | | | | | | | | | | |
| No | 1.00 | (ref) | | 1.00 | (ref) | | 1.00 | (ref) | | 1.00 | (ref) | |
| Yes | 1.74 | 1.38,2.20 | <0.001 | 1.78 | 1.40,2.25 | <0.001 | 1.95 | 1.47,2.58 | <0.001 | 1.91 | 1.42,2.55 | <0.001 |
| **Prior spontaneous abortion** (ref = no) | | | | | | | | | | | | |
| No | 1.00 | (ref) | | 1.00 | (ref) | | 1.00 | (ref) | | 1.00 | (ref) | |
| Yes | 0.74 | 0.66,0.84 | <0.001 | 0.74 | 0.66,0.84 | <0.001 | 0.70 | 0.58,0.84 | <0.001 | 0.70 | 0.58,0.84 | <0.001 |
| **Pregnancy test** (ref = no) | | | | | | | | | | | | |
| No | 1.00 | (ref) | | 1.00 | (ref) | | 1.00 | (ref) | | 1.00 | (ref) | |
| Yes | 0.47 | 0.32,0.69 | <0.001 | 0.48 | 0.33,0.71 | <0.001 | 0.42 | 0.29,0.61 | <0.001 | 0.41 | 0.27,0.61 | <0.001 |
| **Health center proximity** (ref = resides out of district) | | | | | | | | | | | | |
| Resides outside of district | 1.00 | (ref) | | 1.00 | (ref) | | 1.00 | (ref) | | 1.00 | (ref) | |
| Resides within district | 1.56 | 1.08,2.25 | 0.024 | 1.55 | 1.08,2.22 | 0.017 | 1.72 | 1.04,2.83 | 0.034 | 1.73 | 1.04,2.87 | 0.034 |
| Resides within catchment area | 1.55 | 1.06,2.26 | 0.017 | 1.53 | 1.05,2.23 | 0.026 | 1.82 | 1.28,2.58 | <0.001 | 1.84 | 1.28,2.66 | <0.001 |
| **CHW recommended health center visit** (ref = no) | | | | | | | | | | | | |
| No | 1.00 | (ref) | | | | | 1.00 | (ref) | | | | |
| Yes | 1.03 | 0.82,1.30 | 0.793 | | | | 0.97 | 0.74,1.27 | 0.826 | | | |
| **Food insecurity** (ref = no) | | | | | | | | | | | | |
| No | 1.00 | (ref) | | | | | 1.00 | (ref) | | | | |
| Yes | 0.90 | 0.71,1.13 | 0.356 | | | | 1.12 | 0.82,1.54 | 0.457 | | | |
| **Open communication with partner** (ref = no) | | | | | | | | | | | | |
| No | 1.00 | (ref) | | 1.00 | (ref) | | | | | | | |
| Yes | 0.72 | 0.56,0.93 | 0.011 | 0.72 | 0.56,0.93 | 0.011 | | | | | | |

Various factors may be at play, including the stigma and social isolation associated with teenage pregnancy [43]. Our results suggest that targeting adolescents and those with lower educational attainment may be particularly high-yield in improving rates of early ANC attendance and should be a priority for CHW outreach efforts.

**Table 4. Obstetric risk factors identified at ANC-1 associated with delayed antenatal care in univariate models.**

| Characteristics | ≥ 16 weeks gestational age | | | | | ≥ 24 weeks gestational age | | | | |
|---|---|---|---|---|---|---|---|---|---|---|
| | N | % | OR | 95% CI | P-value | N | % | OR | 95% CI | P-value |
| None | 5740 | 91.9 | 0.74 | 0.63, 0.86 | <0.001 | 2320 | 91.8 | 0.85 | 0.72, 1.01 | 0.064 |
| Anemia | 154 | 2.5 | 0.99 | 0.77, 1.28 | 0.953 | 52 | 2.1 | 0.78 | 0.57, 1.06 | 0.123 |
| Proteinuria | 981 | 15.7 | 0.62 | 0.56, 0.68 | <0.001 | 345 | 13.7 | 0.62 | 0.55, 0.71 | <0.001 |
| Hypertension | 16 | 0.3 | 5.11 | 1.45, 32.4 | 0.030 | 5 | 0.2 | 1.17 | 0.38, 3.11 | 0.761 |
| HIV | 135 | 2.2 | 0.73 | 0.53, 1.00 | 0.049 | 32 | 1.3 | 0.77 | 0.51, 1.12 | 0.180 |
| MUAC <21cm | 135 | 2.2 | 1.14 | 0.86, 1.52 | 0.378 | 51 | 2.0 | 0.97 | 0.70, 1.33 | 0.860 |
| Smoking | 42 | 0.7 | 0.76 | 0.49, 1.20 | 0.241 | 12 | 0.5 | 0.56 | 0.29, 1.00 | 0.067 |
| Alcohol use | 1006 | 16.1 | 1.16 | 1.04, 1.29 | 0.010 | 423 | 16.7 | 1.15 | 1.01, 1.29 | 0.028 |
| Multiple births | 220 | 3.5 | 1.90 | 1.47, 2.50 | <0.001 | 106 | 4.2 | 1.74 | 1.36, 2.21 | <0.001 |

Participants in our cohort who received a pregnancy test were less likely to delay ANC-1. Pregnancy testing has been shown to increase early attendance at ANC in multiple low-resource contexts [41, 44, 45], which may be due to earlier pregnancy discovery [18, 19, 46]. Although our analysis did not specifically assess whether patients delayed care due to late discovery, our results suggest that access to pregnancy testing may increase early attendance. Home pregnancy testing has been successfully incorporated into CHW outreach efforts in other low-resource settings, and has been linked to earlier ANC attendance [45, 47]. In Rwanda, where a robust CHW outreach program already exists but access to home pregnancy testing is limited, incorporating pregnancy testing in regular outreach efforts may be an effective strategy to increase early attendance at ANC.

We anticipated that participants who sought care within their home district would be less likely to delay ANC-1. The current ANC care system in Rwanda was restructured in the past two decades, with the goal of providing the majority of ANC services at community health centers [48]. Our results confirm that those who attended their first ANC-1 visit at their local health center were less likely to delay care. While local community health centers improve initial access to care, patients with high-risk conditions identified at ANC-1 may still be required to travel to a district hospital for ongoing ANC [48]. Patients with complex pregnancy-related risk factors, who benefit the most from consistent prenatal care, many face the greatest barriers to accessing services due to the distance of referral centers. Further research is warranted to explore the impact of referral to a district hospital on continued engagement with ANC services for women with identified complications.

**Table 5. Obstetric risk factors identified at ANC-1 associated with parity in univariate models.**

| Characteristics | Nulliparous | | Multiparous | | | | |
|---|---|---|---|---|---|---|---|
| | N | % | N | % | OR | 95% CI | P-value |
| None | 2774 | 97.3 | 6708 | 90.9 | 0.28 | 0.22, 0.35 | <0.001 |
| Anemia | 82 | 2.9 | 171 | 2.3 | 0.80 | 0.62, 1.05 | 0.376 |
| Proteinuria | 503 | 17.6 | 1400 | 19.0 | 1.09 | 0.98, 1.22 | 0.734 |
| Hypertension | 3 | 0.1 | 15 | 0.2 | 1.93 | 0.64, 8.35 | 0.297 |
| HIV | 33 | 1.2 | 126 | 1.7 | 1.48 | 1.02, 2.22 | 0.045 |
| MUAC <21cm | 57 | 2.0 | 154 | 2.1 | 1.04 | 0.77, 1.43 | 0.780 |
| Smoking | 19 | 0.7 | 58 | 0.8 | 1.18 | 0.72, 2.04 | 0.531 |
| Alcohol use | 311 | 10.9 | 1262 | 17.1 | 1.68 | 1.48, 1.93 | <0.001 |
| Multiple births | 2 | 0.1 | 293 | 4.0 | 58.9 | 18.9, 335 | <0.001 |

Multiparous patients in our cohort were more likely to present late to care. Reasons for delaying ANC among multiparous patients is complex, although some evidence suggests that women delay ANC due to a history of uncomplicated pregnancies and the perception that their current pregnancy will be similarly uncomplicated [24–26]. Given this emerging evidence, we expected that multiparous women who had experienced an obstetric complication would be less likely to delay ANC. The results of our analysis were mixed. While women who reported a preterm birth or low birthweight delivery were less likely to delay ANC, those with a history of stillbirth or neonatal mortality delayed care at rates similar to those without a history of these adverse outcomes. Preterm and low birthweight deliveries generally result in neonatal hospitalization, while a stillbirth or neonatal death may not. Past experience with the healthcare system may explain earlier attendance among patients with a history of preterm or low birthweight deliveries. Those with a history of a spontaneous abortion were also less likely to delay ANC, which may be due to increased motivation to prevent future pregnancy losses [18]. In an effort to increase attendance at ANC, educational outreach should stress that early monitoring and interventions during pregnancy may decrease adverse neonatal outcomes, including infant death.

Surprisingly, we found that women who contributed to their household's income were more likely to delay ANC-1. Our results stand in contrast to other investigations that have found that women in paid employment positions utilize ANC at higher rates than those who are unemployed [19]. One possible explanation for our results is that mothers may prioritize the immediate financial needs of their families over early attendance at ANC in this low-resource context. Additional studies are needed to better understand the impact of maternal employment on ANC utilization in Rwanda.

Our results also demonstrated striking disparities in ANC attendance between districts. Nyamasheke had the lowest proportion of women presenting after 16 weeks at 45.1%, while Rubavu had the highest proportion at 75.3%. Similarly, only 13.5% of women in Nyamasheke presented after 24 weeks GA, while 40.6% of women in Rubavu presented after 24 weeks. In addition to their CHW program, a pillar of Rwanda's maternal health strategy is their health center incentivization scheme, which rewards facilities based on the number of patients who attend ANC-1 before 16 weeks and complete at least 4 ANC visits [49]. These types of incentive schemes may affect how local health centers interact with communities and work to encourage women in their communities to participate in care [50]. Further investigation into the successes of high-performing districts may yield additional insights into effective strategies for encouraging women to seek early antenatal care.

This investigation also examined differences in the obstetric risk factors identified at participants' first ANC visit. Given that the risk of developing certain obstetric risk factors increases throughout pregnancy, we expected that presenting late to care would increase the likelihood of having an obstetric complication identified at ANC-1. Our analysis confirmed these results, and we found that women presenting after 16 weeks GA were more likely to have an obstetric risk factor identified at ANC-1. While women who present early to care may go on to develop a risk factor later on in pregnancy, early and frequent access to care promotes timely identification of new obstetric risks. We also examined the effect of parity on the risk factors identified at ANC-1 and found that multiparous women presented to ANC-1 with similar risk factors to nulliparous women. In the setting of emerging evidence that multiparous women may delay ANC due to the belief that their pregnancies are lower-risk [46], these results raise concern that delays in care among multiparous patients may result in later identification of important obstetrical risk factors.

Despite conducting data strengthening activities, our investigation found lower than expected rates of obstetric risk factors in our study population. Some of the most important

causes of maternal and neonatal morbidity and mortality including hypertension [51–53], HIV positive status [54], and malnutrition (MUAC) [55, 56] were below expected prevalence. Other important screening results, including diabetes and syphilis, were systematically missing from the health record and thus were excluded from our final analysis. Low MUAC represents a particularly interesting example of an underreported risk factor, as maternal malnutrition is highly visible during pregnancy and should be easily identified both by community-based CHWs and ANC providers. These results demonstrate weaknesses in the identification and documentation of modifiable obstetrics risk factors, one of the most important functions of ANC.

In the context of recent calls from the WHO to double the number of ANC contacts, our results raise concerns about the quality of care within the current four visit system. One Rwandan study found that basic screening and prophylactic care was missing in up to 15% of ANC patient charts. This study also reported large gaps in provider knowledge, including the ability to identify key pregnancy-related conditions that would require urgent referral to a higher level facility [57]. While Rwanda has a stronger healthcare system than many of its regional peers, our data show there is room for improvement in terms of capturing important risk factors in the antenatal period. A few studies have demonstrated that increasing the number of ANC visits a woman attends does not necessarily result in improved pregnancy outcomes, if important screening and educational activities are missed [33, 58]. If Rwanda moves towards an eight-visit system, using additional contacts as an opportunity to enhance the delivery of educational activities and screening through provider trainings and standardized protocols will be important for improving maternal and neonatal outcomes.

## Limitations

Since this investigation was a secondary data analysis and some variables had limited completion rates, we were unable to assess the impact of certain factors such as health insurance or family wealth on ANC attendance. Even though Rwanda has the highest insurance rates in sub-Saharan African (over 90%) and the majority of ANC services are provided within community health centers to decrease the financial burden of travel, financial concerns may impact ANC attendance for the small proportion of uninsured women [59, 60]. In addition, certain variables such as marital status and pregnancy intention were not collected in the primary dataset, and thus we were unable to include them in our analysis. Due to poor data quality, we were unable to assess important obstetric risk factors typically identified at ANC, including diabetes and syphilis. We were also unable include infection with malaria, as malaria testing is not required at ANC-1 and many health systems did not screen for malaria at their prenatal visits. While data strengthening exercises were conducted at each participating health center in an attempt to improve variable completeness, a parallel system of data collection would have strengthened this study and ensured a more accurate capture of clinical information.

## Conclusions

Early attendance at ANC is important for the early identification and management of obstetric risk factors. In Rwanda, increasing access to ANC is one of the leading health sector priorities. While the country has been successful in achieving near universal access to at least one ANC visit, our study found that the majority of Rwandan women continue to present late to care. Through this investigation, we identified multiple sociodemographic predictors of late presentation to ANC and proposed a few possible interventions to promote earlier ANC attendance. Building upon Rwanda's robust CHW network to administer pregnancy testing is one example and may prove to be a low-cost way to prompt earlier attendance at ANC clinics. CHW

outreach has been very successful in increasing overall attendance at ANC and targeting their efforts towards those most at risk of delaying ANC is another strategy that may prompt earlier care initiation. Existing efforts by the Rwandan government to locate ANC services within communities, achieve universal health insurance, and leverage CHWs to link communities to care are important steps towards improving early ANC attendance. Our results also demonstrated gaps in the identification and documentation of important obstetric risk factors at ANC-1 visits. Health centers should focus on enhancing the delivery of educational activities and improving antenatal screening through standardized provider trainings and protocols.

## Supporting information

**S1 File. Participant enrollment survey.**
(DOCX)

## Acknowledgments

We are grateful to the PTBi-Rwanda field coordinators who supported the administration of our surveys and collection of clinical data from our health centers. We thank the PTBi-Rwanda project managers and staff, who contributed to data cleaning and organization for the greater East Africa Preterm Birth Initiative.

## Author Contributions

**Conceptualization:** Christina N. Schmidt, Elizabeth Butrick, Dilys Walker.

**Data curation:** Nathalie Mulindahabi.

**Formal analysis:** Christina N. Schmidt.

**Methodology:** Christina N. Schmidt, Elizabeth Butrick.

**Supervision:** Elizabeth Butrick, Dilys Walker.

**Visualization:** Christina N. Schmidt.

**Writing – original draft:** Christina N. Schmidt.

**Writing – review & editing:** Elizabeth Butrick, Sabine Musange, Nathalie Mulindahabi, Dilys Walker.

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
