## [Decision Letter · Decision Letter 0]

18 Nov 2020

PONE-D-20-22930

Towards stronger antenatal care: understanding predictors of late presentation to antenatal services and implications for obstetric risk management in Rwanda

PLOS ONE

Dear Dr. Schmidt,

Thank you for submitting your manuscript to PLOS ONE. After careful consideration, we feel that it has merit but does not fully meet PLOS ONE’s publication criteria as it currently stands. Therefore, we invite you to submit a revised version of the manuscript that addresses the points raised during the review process.

Please see the comments from the reviewers below. The reviewers have requested further elaboration on the statistical analysis, how the methodology ties in with the aims, justification for specific choices in the methodology and analysis, as well as clarification on certain specific points.

We look forward to receiving your revised manuscript.

Kind regards,

Hanna Landenmark

Associate Editor

PLOS ONE

Journal Requirements:

2. "You indicated that you had ethical approval for your study. In your Methods section, please ensure you have also stated whether you obtained consent from parents or guardians of the minors included in the study or whether the research ethics committee or IRB specifically waived the need for their consent.

4. Please ensure you have included the registration number for the clinical trial referenced in the manuscript.

Reviewers' comments:

Reviewer's Responses to Questions

**Comments to the Author**

1. Is the manuscript technically sound, and do the data support the conclusions?

Reviewer #1: No

Reviewer #2: No

Reviewer #3: Partly

Reviewer #4: Yes

Reviewer #5: Yes

2. Has the statistical analysis been performed appropriately and rigorously? 

Reviewer #1: No

Reviewer #2: No

Reviewer #3: No

Reviewer #4: No

Reviewer #5: Yes

3. Have the authors made all data underlying the findings in their manuscript fully available?

Reviewer #1: Yes

Reviewer #2: Yes

Reviewer #3: No

Reviewer #4: Yes

Reviewer #5: Yes

4. Is the manuscript presented in an intelligible fashion and written in standard English?

Reviewer #1: Yes

Reviewer #2: Yes

Reviewer #3: Yes

Reviewer #4: Yes

Reviewer #5: Yes

5. Review Comments to the Author

Reviewer #1: A secondary analysis was conducted with the aim of identifying factors associated with delayed initiation of antenatal care (ANC) and comparing the risks between those presenting early and late for care. Adjusting for clusters, backwards stepwise regression analysis were used to identify predictors of first ANC visit. The statistical methods are vague; therefore, the conclusions are unclear.

Major revision:

It seems that either logistic regression models should be used when predicting the dichotomies of ≥ 16 weeks or ≥ 24 weeks or linear regression models to predict the continuous value for week of ANC-1. Additionally, in the backward selection model it is common to include candidate predictors with univariate p-values ≤ 0.10.

Minor revisions:

Line 179: Chi-square tests are used to show association. Use logistic or linear regression models for prediction, even for univariate tests.

Reviewer #2: Abstract

The results are confusing. The study set out to assess factors associated with delayed initiation of the first antenatal visit, in lines 31-33. However, it is reporting factors associated with early initiation of antenatal care in part of the results section, rather than factors associated with late initiation of antenatal care, in lines 41-47.

• Educational attainment, prior spontaneous abortion and pregnancy testing were associated with earlier ANC-1 (p<0.001).

• Women with a prior preterm (p=0.024) or low birthweight (p=0.023) delivery were more likely to present before 16 weeks GA, while history of stillbirth or neonatal death were not associated with timing of ANC-1.

• Women presenting before 16 weeks GA were more likely to have no obstetric risk factors identified at ANC-1 (p<0.001).

In lines 49-54, the authors present a discussion, but do not have a clear conclusion. Also, the authors do not indicate how awareness of the importance of antenatal care would be increased. They also do not indicate how health system strengthening may be done to identify the suggested risk factors. They do not seem to have a conclusion. They state:

• Outreach efforts should focus on increasing mothers’ awareness of the benefits of ANC and emphasize the importance of early identification and management of pregnancy-related complications.

• While fewer risk factors were identified among women presenting less than 16 weeks, lower than expected rates were identified in the study population overall.

Main article

In lines 65-71, the authors state that many of the complications that lead to adverse outcomes can be maternal or neonatal outcomes can be prevented through early antenatal care, and mention hemorrhage as one of the factors, yet antenatal care may not mitigate hemorrhage-related maternal morbidity and mortality. Similarly, they suggest that antenatal care may reduce infection-related neonatal morbidity and mortality without any suggestion of how this can be achieved. This to me seems speculative, considering that most neonatal infection follow peripartum or postnatal complications.

In lines 84-93, the authors should provide an explanation of how many of the suggested factors lead to late presentation of antenatal care.

Methods

In lines 158-162, the authors need to indicate the definitions used for anemia, hypertension, proteinuria and diabetes, as well as how these variables were measured. Is it present/absent or a definite cut off?

In lines 171-178, the data analysis conducted is not clear. At analysis, the authors state:

Variables that were significantly associated with delayed presentation to ANC at the 172

=0.05 level were retained for multivariable model building. This is too strict and erroneous, as

they should have included all variables with a p-value of less than 0.2 or even those with clinical

significance, such as complications in a prior pregnancy. The backward and stepwise method of

variable selection is also not clear.

Discussion

The discussion does not provide an explanation of how the suggested factors operate, for example educational attainement, parity, prior complications etc.

It is also not clear how parity as a moderator was assesed, when you view the results presented, as the stratified analysies compared primiparous and multiparous, rather than parity as an independent variable.

In lines 331-335, the authors state that : This investigation also examined differences in the obstetric risk factors identified at participants’ first ANC visit. Women presenting before 16 weeks GA were more likely to have no identified risk factors (“none”). This may be partially due to the fact that some obstetric risk factors, such as multiple births or hypertension, may be identified at a higher rate at later gestational ages. However, the focus of their stated objective was to assess factors associated with late antenatal attendance. Also, their explanation of of lack of identifiable risk factor is not clear to me.

The authors add, in lines 338-340, that: multiparous patients do not appear to be at lower

risk of pregnancy-related complications. Multiparous women should be encouraged to continue

to attend ANC, due to the risk of delayed identification of modifiable risk factors. This also is not clear to me.

The authors should comment, in the discussion, on the quality of antenatal care, that is activities conducted per visit, rather than the number and timing of antenatal care visits, and the influence of this on pregnancy outcomes (maternal and mneonatal outcomes).

Reviewer #3: 1. Abstract

On line 41‘’Earning income’’: not clear

Line 48, use conclusion instead of discussion

2. Introduction

On page 6 line 112 ‘‘this analysis identified predictors of late (≥16 weeks GA)…’’. Why did you use 16 weeks of gestational age as a cut-off pint? WHO recommends pregnant women have to start their first ANC booking within the first 12 weeks of gestational age. See the following references.

World Health Organization. WHO recommendations on antenatal care for a positive pregnancy experience [Internet]. Geneva, Switzerland; 2016. Available from: ttps://apps.who.int/iris/bitstream/10665/250796/.../9789241549912-eng.pdf

Tuncalp Ӧ, Pena-Rosas J, Lawrie T, Bucagu M, Oladapo O, Portela A, et al. WHO recommendations on antenatal care for a positive pregnancy experience — going beyond survival. BJOG. 2017;124:860–2.

3. Methods

From line 132-133, ‘‘only participants with completed ANC care records were included in the final analysis.’’ If this is the case, is it representative?

4. Analysis

Data analysis is not well addressed and robust

From line 165-166, ‘‘Chi-squared tests were performed to identify variables

significantly associated with late presentation to ANC.’’ why you preferred Chi-squared tests than odds ratios?

Not clear on the selection of measures of association used. For instance, Chi-squared tests vs. Odds ratios.

5. Ethical consideration

Line 198, ‘‘Written consent was obtained from all participants.’’ How did you secure it? Are all read and write to give you written consent?

6. Discussion

Lacks author argument

7. Limitations

Line 370, ‘‘we found that many clinical variables had poor quality.’’ many clinical variables specifically………had poor quality

Reviewer #4: Towards stronger antenatal care: understanding predictors of late

presentation to antenatal services and implications for obstetric risk

management in Rwanda

This study is of Public health relevance. The authors identified factors associated with delayed initiation of ANC and describe differences in the obstetric risks identified at the first ANC (ANC-1) visit between women presenting early and late to care.

I have the following comments

1. L36 & L172…authors claimed to have used “Cluster-robust standard error models”. This is a model estimation method and not a model in itself. In essence, you used logistic regression and controlled for the clustering effect of the health centres

2. What was the level of significance adopted?

3. What was the main study outcome and variables explored?

4. L127 …what were the controls

5. What is the rationale for grouping women aged 20 to 34 together? This is a critical age group where most pregnancy occurred. The behaviour of young adults (20-24 years) are very distinct from those aged 25-34 years. Authors should reanalyse

6. Important variables such as Type of marriage (monogamy/ polygamy) was, autonomy, decision making power by the women were excluded

7. Table 3: Why present both coefficient estimates and odds ratio? Delete the two columns on “Estimate (SE)”. It is redundant.

8. Did you adjust for multicollinearity among the explanatory variables?

9. Did authors assess the relationship between individual explanatory variables before choosing them as candidate variables in the multiple regression model?

10. It is usual to first do this assessment (bivariate regression) and include only the variables that were significant at a specified level (say 10 or 20%) before you run the multiple regression. The outputs of the bivariate regression are the odds ratio, sometimes called crude odds ratio (COR) while those from the multiple regression are called adjusted odds ratio (AOR)

11. Tables 4 and 5 should precede Table 3

12. Why were the risk factors in Tables 4 and 5 excluded from the regression model in Table 3? Were they insignificant?

13. There are typos and grammar issues

14. Authors should make the public health importance of this study clearer

Reviewer #5: Reviewer's report:

Initiating antenatal care (ANC) in the first trimester and identifying obstetric risk factors early are crucial for improving birth outcomes. Also, service providers’ knowledge of predictors of initiation of ANC has implications for obstetric risk management. This can help improve quality of care. However, English language needs some input; a few have provided.

General comments

1. The abstract convey what is in the manuscript

2. The data is sound.

3. The method section of the manuscript requires a major revision

4. The tables are genuine and well described

5. Some parts of the discussions needs revision to situate especially contrasting results in academic discourse

6. Conclusions reflect the results

I thus, recommend that the manuscript be accepted and the authors asked to revise the manuscript before publication.

Abstract

Word count =306

1. The title: Towards stronger antenatal care: understanding predictors of late presentation to antenatal services and implications for obstetric risk management in Rwanda

The title needs revision to cover early initiation of ANC (<16 weeks) since it is part of the analysis as shown in Table 1 and the conclusion

Introduction

1. P. 5, line 89: insert ‘as’ for the sentence read such as the practice of disclosing pregnancy after the second trimester.

2. P. 5, line 90-91: I suggest ‘poor experiences’ be replaced with ‘negative experiences’

3. P. 7, line 119-121: The authors should avoid repeating words sentences. The second ‘how’ in the sentence. I suggest: …, and inform messages tailored to multiparious and nulliparous women.

4. P. 7, 133: … completed ANC care… the ‘care’ is repetition and should be deleted.

Results

5. P. 10, line 203 and 206-208: By conversion, a sentence should not begin with a figure. I suggest the sentences should be re-written.

6. P. 15 and 16, Tables 1, 2, 3 and 4 and 5: The ‘p-values’. The ‘p’ should be capitalised as ‘P-values’

7. P. 16: Idetified is not necessary in Table 5

8. All the information in brackets in Tables 1, 2, 3 and 4 (≥ 16 weeks GA) and very late (≥ 24 weeks GA) is not necessary. They are already difined in the text.

9. I expect the authors to emphasise finding regarding the benefits of initiating ANC < 16 weeks.

Discussion

10. P. 17, line 276: Despite ongoing to increase early ANC attendance, that the .... ‘that’ should be deleted.

11. P. 17, 283-287: I suggest the authors look for evidence to explain and support their assertion that contribution to household income was a stronger predictor of delayed ANC-1. Also, could you please emphasise what your study adds?

12. P. 18, line 295: More effective strategy increase .... There should be ‘to’ as: ... strategy to increase ...

13. P. 18, line 300-302: However, barriers to continuing ANC care may emerge for women for whom ANC-1. The sentence should be: However, barriers to continuing ANC care may emerge for women for whom ANC-1. I suggest ‘ANC’ be deleted and ‘for whom’ replaced with ‘whose’

14. P. 19, line 333: why put ‘none’ into brackets. The sentence clear without ‘none’

15. P. 20, line 342: Despite conducting data strengthening activities, our investigation found lower than expected rates of obstetric risk factors were identified in the study population overall. ‘were identified’ is a repetition so need to be deleted.

16. P. 20, line 345-346: The sentence needs to be re-written to make comprehension easy for readers. I suggest: Other important screening results including diabetes and syphilis were systematically missing from the health record and thus were excluded from our final analysis.

17. P. 20, line 351: ... function of ANC care. ‘care’ is repetition and should be deleted.

18. P. 20, line 357: Delete the second ‘in’ in the sentence. i.e. This study also reported large gaps in provider knowledge, including the ability to …

Conclusion

19. The conclusion should also emphasise the benefits and predictors of early ANC attendance to inform interventions.

20. The conclusion early intervention is too unspecific as this vary in different cultures

21. The recommendation that community-based pregnancy testing could increase early pregnancy discovery and prompt earlier attendance at ANC clinics can be contested as the data does not show that late attendance is attributed to not being aware

6. PLOS authors have the option to publish the peer review history of their article (what does this mean?). If published, this will include your full peer review and any attached files.

Reviewer #1: No

Reviewer #2: **Yes: **Daniel Kabonge Kaye

Reviewer #3: **Yes: **Tufa Kolola Huluka

Reviewer #4: No

Reviewer #5: No

---

## [Author Response · Author response to Decision Letter 0]

28 Dec 2020

Please see our "Response to reviewers" letter, in which we respond to the feedback that we received from our reviewers. We appreciate the opportunity to revise and resubmit this manuscript.

---

## [Decision Letter · Decision Letter 1]

31 Mar 2021

PONE-D-20-22930R1

Towards stronger antenatal care: Understanding predictors of late presentation to antenatal services and implications for obstetric risk management in Rwanda

PLOS ONE

Dear Dr. Schmidt,

Thank you for submitting your manuscript to PLOS ONE. After careful consideration, we feel that it has merit but does not fully meet PLOS ONE’s publication criteria as it currently stands. Therefore, we invite you to submit a revised version of the manuscript that addresses the points raised during the review process.

We look forward to receiving your revised manuscript.

Kind regards,

Adeniyi Francis Fagbamigbe, Ph.D

Academic Editor

PLOS ONE

Journal Requirements:

Reviewers' comments:

Reviewer's Responses to Questions

**Comments to the Author**

1. If the authors have adequately addressed your comments raised in a previous round of review and you feel that this manuscript is now acceptable for publication, you may indicate that here to bypass the “Comments to the Author” section, enter your conflict of interest statement in the “Confidential to Editor” section, and submit your "Accept" recommendation.

Reviewer #1: (No Response)

Reviewer #3: (No Response)

Reviewer #5: (No Response)

2. Is the manuscript technically sound, and do the data support the conclusions?

Reviewer #1: No

Reviewer #3: Yes

Reviewer #5: Yes

3. Has the statistical analysis been performed appropriately and rigorously? 

Reviewer #1: No

Reviewer #3: Yes

Reviewer #5: Yes

4. Have the authors made all data underlying the findings in their manuscript fully available?

Reviewer #1: Yes

Reviewer #3: Yes

Reviewer #5: Yes

5. Is the manuscript presented in an intelligible fashion and written in standard English?

Reviewer #1: Yes

Reviewer #3: Yes

Reviewer #5: Yes

6. Review Comments to the Author

Reviewer #1: Major Revision:

Tables 1 and 3: In the logistic model each characteristic will have only one p-value associated with it. In other words, p-values for each level of a characteristic should not be provided. Consider consulting a qualified statistician for assistance with reporting the results from logistic regression modeling.

Minor Revisions:

1- Express p-values more precisely than “p <0.05” and "p<0.01".

2- Line 228: Clarify the meaning of alpha=0.05.

3- The standard statistical term for average is mean.

4- Line 246: Provide a measure of dispersion for the mean of GA at presentation.

5- Clarify if the results presented in Tables 4 and 5 are based on univariate or multivariate models.

6- Spell out a number when it occurs at the beginning of a sentence.

Note: Line numbers refer to those in the tracked changes version of revision 1.

Reviewer #3: I appreciate the authors' effort because I have observed a significant improvement they made into this paper at this stage. The authors have also addressed the majority of my concern during their revision. But, still, I am not convinced in some cases.

1.For instance, regarding cutoff for “late” presentation, you referenced the cutoff used by the Rwandan Ministry of Health which is not standard or consistent with the cutoff used elsewhere. When I see published articles on this issue, the majority of articles published since 2018 used >12 weeks GA as “late” presentation. My concern here is that how yours compared with results obtained by this cutoff (>12 weeks GA)? In Rwanda, it may be OK.

2.Regarding the selection of measures of association, I am not clear with the reason why did you prefer reporting p-value than odds ratio? As you know p-value conveys a little information regarding the strength and direction of association compared with odds ratio. Additionally, even your p-value report is not consistent throughout. For instance, p<0.05, p<0.001, p=0.02, p=0.03,etc. When you report the p-value, report its exact value.

Reviewer #5: The manuscript is much improved, the text flows much better and almost all the issues raised have been addressed and acceptable for publiction.

However, there are a few spelling and grammar mistakes (eg missing words, incorrect verb conjugation, missing articles etc) Examples

P.3 line 42, by conversion a sentence should not begin with a figure

P.5 line 83, ‘greater’ not necessary in the sentence.

P.6 line 101, "of" is missing in …parity and history of uncomplicated …

P23 line 437 the sentence should start with ‘The’ and replace the second ‘CHW’ with ‘their’ to avoid repetition.

P23 line 441 replace ‘demonstrated’ with ‘demonstrate’

P.22 line 402-404

This sentence seem to belittle the WHO’s call for more ANC visits. The eight ANC visit required is expected to ensure that all the necessary education activities and screening is carried out more thoroughly. Though arguably the intention is to ensure that even if a pregnant woman misses some ANC appointments, at least she will benefit from the education activities and screening than having four with some deciding to go only once or twice.

P.23 line 442-445.

You did not assess quality of care at ANC clinics. Why let your concluding statement be on quality of care. As stated above I suggest that being the last sentence, it should rather echo WHOs’ call for increasing number of ANC visits and the CHW outreach programme used to vigorously sensitise pregnant women and communities on the need to initiate ANC in the first trimester. This is likely to increase the number of ANC visits and improve pregnancy and birth outcomes as shown by your results.

7. PLOS authors have the option to publish the peer review history of their article (what does this mean?). If published, this will include your full peer review and any attached files.

Reviewer #1: No

Reviewer #3: **Yes: **Tufa Kolola

Reviewer #5: No

---

## [Author Response · Author response to Decision Letter 1]

15 May 2021

Please see the end of our "Response to reviewers" letter, in which we respond to the reviewer and editor comments in detail.

---

## [Decision Letter · Decision Letter 2]

2 Jun 2021

PONE-D-20-22930R2

Towards stronger antenatal care: Understanding predictors of late presentation to antenatal services and implications for obstetric risk management in Rwanda

PLOS ONE

Dear Dr. Schmidt,

Thank you for submitting your manuscript to PLOS ONE. After careful consideration, we feel that it has merit but does not fully meet PLOS ONE’s publication criteria as it currently stands. Therefore, we invite you to submit a revised version of the manuscript that addresses the points raised during the review process.

We look forward to receiving your revised manuscript.

Kind regards,

Adeniyi Francis Fagbamigbe, Ph.D

Academic Editor

PLOS ONE

Journal Requirements:

Additional Editor Comments (if provided):

I recommend minor revisions. The Reviewer 1 has raised important statistical issue amongst others. This particular issue has been raised previously but was not attended to

Table 3: Provide the overall p-values for each factor and remove the p-values associated with each level.

In logistic regression and any other regression involving categorical variables, the variable can not have a p-value, rather, all its categories except the reference category should have p-values.

You have to specify the reference category for each variable, the odds ratio for the reference categories is always 1.oo and it wont have a p-value. For example in the Age variable, if 15-19 is the reference, then age wont have p-value, 15to 19 wont have pp-value but other categories must have p-values.

Kindly note that the editorial may have no other choice to reject the manuscript if you fail to correct this in this round

Reviewers' comments:

Reviewer's Responses to Questions

**Comments to the Author**

1. If the authors have adequately addressed your comments raised in a previous round of review and you feel that this manuscript is now acceptable for publication, you may indicate that here to bypass the “Comments to the Author” section, enter your conflict of interest statement in the “Confidential to Editor” section, and submit your "Accept" recommendation.

Reviewer #1: (No Response)

Reviewer #3: All comments have been addressed

2. Is the manuscript technically sound, and do the data support the conclusions?

Reviewer #1: Yes

Reviewer #3: Yes

3. Has the statistical analysis been performed appropriately and rigorously? 

Reviewer #1: No

Reviewer #3: Yes

4. Have the authors made all data underlying the findings in their manuscript fully available?

Reviewer #1: Yes

Reviewer #3: Yes

5. Is the manuscript presented in an intelligible fashion and written in standard English?

Reviewer #1: Yes

Reviewer #3: Yes

6. Review Comments to the Author

Reviewer #1: 1- Line 218: Clarify alpha=0.05. When factors are tested in models, p-values rather than alpha levels are generated.

2- Table 1: The results of the logistic regression models are not summarized in a sensible fashion. Based on the authors' explanation, I determined that the OR for age of 1.01 is associated with a one-year increment in age. Typically a 5 or 10 year increment in age is commonly provided when age is modeled continuously. Additionally, if the authors prefer to model age as a categorical factor, then the reference age group must be identified so the ORs have meaning. The reference group is the group to which all other groups are compared and is typically designated as the lowest or highest category. This table requires revisions, and I suggest using categorical representations of the factors and removing the continuous representation. My suggestion of providing an overall p-value and removing the level p-values still stands because a reader can determine which groups differ by carefully examining the 95% CI for the OR. If the CI does not contain 1, it differs significantly from the reference group.

3- Table 3: Provide the overall p-values for each factor and remove the p-values associated with each level.

Reviewer #3: (No Response)

7. PLOS authors have the option to publish the peer review history of their article (what does this mean?). If published, this will include your full peer review and any attached files.

Reviewer #1: No

Reviewer #3: **Yes: **Tufa Kolola

---

## [Author Response · Author response to Decision Letter 2]

17 Jul 2021

July 15th, 2021

Dr. Adeniyi Francis Fagbamigbe

Dear Dr. Fagbamigbe, 

Thank you for the opportunity to continue to revise and resubmit our manuscript PONE-D-20-22930-R3, entitled “Towards stronger antenatal care: Understanding predictors of late presentation to antenatal services and implications for obstetric risk management in Rwanda.”

We appreciate the reviewer’s feedback and have responded to each of the suggestions raised during this round of reviews. We have attempted to address each point, taking in to account the feedback from our reviewer and from you, our Editor. We are invested in publishing this work with PLOS ONE, and are more than happy to make any additional changes, as necessary.

In this document we have included a response to each comment the reviewers made. In our response to reviewers, the reviewers’ comments are numbered, and our responses follow below prefaced by “Author response.” Corresponding changes are highlighted in the manuscript text in the revised file. 

Thank you again for your continued consideration of our manuscript. We look forward to hearing from you in due time regarding our submission and to respond to any further questions and comments you may have.

Sincerely, 

Christina Schmidt (corresponding author) 

Doctor of Medicine Candidate, School of Medicine 

University of California San Francisco 

Email: christina.schmidt@ucsf.edu

Phone: +1 503 703 4953

 

Response to Editor’s comments: 

I recommend minor revisions. The Reviewer 1 has raised important statistical issue amongst others. This particular issue has been raised previously but was not attended to

E1.1: Table 3: Provide the overall p-values for each factor and remove the p-values associated with each level.

In logistic regression and any other regression involving categorical variables, the variable can not have a p-value, rather, all its categories except the reference category should have p-values.

You have to specify the reference category for each variable, the odds ratio for the reference categories is always 1.oo and it wont have a p-value. For example in the Age variable, if 15-19 is the reference, then age wont have p-value, 15to 19 wont have pp-value but other categories must have p-values. Kindly note that the editorial may have no other choice to reject the manuscript if you fail to correct this in this round

Author response: Thank you for this feedback. As you have suggested, in this current revision we do not include a p-value for our reference categories. We previously designated our reference categories with the word “reference” and have adjusted them to instead read OR = 1.00 (ref). For example, for age, we selected 25-29 years as our reference category (as this is the most common age of pregnancy among our study population), and thus have listed OR = 1.00 for this group. We include a p-value and CI for each of the levels that are not the reference group (i.e. age 15-19, 20-24, 30-24 and 35+). We do not include a p-value for the overall variable (i.e. age), as you suggested in your example above. Please let us know if we have misunderstood this feedback, and if you would like us to make any additional adjustments to this table. 

For Table 1, we have adjusted our table according to the feedback we received from Reviewer 1, including a p-values for the overall variable, and ORs with 95% CI for each level without p-values (see R1.2). We defer to our editorial/reviewer team and are more than happy to adjust these tables to present this data in a different way.

Thank you for your ongoing feedback on this manuscript. We have attempted to address the final comments from our reviewers and are happy to make any additional adjustments, as necessary. 

Response to reviewer #1: 

R1.1: Line 218: Clarify alpha=0.05. When factors are tested in models, p-values rather than alpha levels are generated.

Author response: Thank you for this feedback. We have removed this in our revised draft. 

R1.2: Table 1: The results of the logistic regression models are not summarized in a sensible fashion. Based on the authors' explanation, I determined that the OR for age of 1.01 is associated with a one-year increment in age. Typically a 5 or 10 year increment in age is commonly provided when age is modeled continuously. Additionally, if the authors prefer to model age as a categorical factor, then the reference age group must be identified so the ORs have meaning. The reference group is the group to which all other groups are compared and is typically designated as the lowest or highest category. This table requires revisions, and I suggest using categorical representations of the factors and removing the continuous representation. My suggestion of providing an overall p-value and removing the level p-values still stands because a reader can determine which groups differ by carefully examining the 95% CI for the OR. If the CI does not contain 1, it differs significantly from the reference group.

Author response: We appreciate this clarification. We have adjusted our reporting and updated Table 1 accordingly, removing the p-values for each level while maintaining the 95% CIs, and providing an overall p-value. We have identified our reference levels in our updated table to facilitate interpretation of our odds ratios, indicated with OR = 1.0 (ref) in the corresponding row, and a line in the variable header itself. Of note, we selected age 25-29 as our reference level, as this is the most common age of pregnancy among our population. We appreciate your ongoing input on this work and are happy to make any additional adjustments to this table if we have misinterpreted this feedback. 

R1.3: Table 3: Provide the overall p-values for each factor and remove the p-values associated with each level. 

Author response: Thank you for this feedback. Please see our response to the Editor’s comments (E1.1) where we discuss adjustments to Table 3. We are more than happy to make additional changes, as necessary.

---

## [Editor Report · Decision Letter 3]

9 Aug 2021

Towards stronger antenatal care: Understanding predictors of late presentation to antenatal services and implications for obstetric risk management in Rwanda

PONE-D-20-22930R3

Dear Dr. Schmidt,

We’re pleased to inform you that your manuscript has been judged scientifically suitable for publication and will be formally accepted for publication once it meets all outstanding technical requirements.

Kind regards,

Adeniyi Francis Fagbamigbe, Ph.D

Guest Editor

PLOS ONE
---

## [Editor Report · Acceptance letter]

16 Aug 2021

PONE-D-20-22930R3 

Towards stronger antenatal care: Understanding predictors of late presentation to antenatal services and implications for obstetric risk management in Rwanda 

Dear Dr. Schmidt:

I'm pleased to inform you that your manuscript has been deemed suitable for publication in PLOS ONE. Congratulations! Your manuscript is now with our production department. 

Kind regards, 

on behalf of

Dr. Adeniyi Francis Fagbamigbe 

Guest Editor

PLOS ONE